# Quantifying electronic band interactions in van der Waals materials using angle-resolved reflected-electron spectroscopy

Johannes Jobst[1,2], Alexander J.H. van der Torren[1], Eugene E. Krasovskii[3,4,5], Jesse Balgley[2], Cory R. Dean[2], Rudolf M. Tromp[1,6] & Sense Jan van der Molen[1]

High electron mobility is one of graphene's key properties, exploited for applications and fundamental research alike. Highest mobility values are found in heterostructures of graphene and hexagonal boron nitride, which consequently are widely used. However, surprisingly little is known about the interaction between the electronic states of these layered systems. Rather pragmatically, it is assumed that these do not couple significantly. Here we study the unoccupied band structure of graphite, boron nitride and their heterostructures using angle-resolved reflected-electron spectroscopy. We demonstrate that graphene and boron nitride bands do not interact over a wide energy range, despite their very similar dispersions. The method we use can be generally applied to study interactions in van der Waals systems, that is, artificial stacks of layered materials. With this we can quantitatively understand the 'chemistry of layers' by which novel materials are created via electronic coupling between the layers they are composed of.

[1] Huygens-Kamerlingh Onnes Laboratorium, Leiden Institute of Physics, Leiden University, Niels Bohrweg 2, P.O. Box 9504, NL-2300 RA Leiden, The Netherlands. [2] Department of Physics, Columbia University, New York, New York 10027, USA. [3] Departamento de Física de Materiales, Universidad del Pais Vasco UPV/EHU, 20080 San Sebastián/Donostia, Spain. [4] IKERBASQUE, Basque Foundation for Science, E-48013 Bilbao, Spain. [5] Donostia International Physics Center (DIPC), E-20018 San Sebastián, Spain. [6] IBM T.J. Watson Research Center, 1101 Kitchawan Road, P.O. Box 218, Yorktown Heights, New York, New York 10598, USA. Correspondence and requests for materials should be addressed to J.J. (email: jobst@physics.leidenuniv.nl).

Electronic band structure is the key to most properties of crystalline materials. Band structure measurements are therefore widely used to study the subtle interplay of electrons with lattice excitations[1,2] or collective electron phenomena[3]. The bands and their dispersions originate from the quantum overlap between the electronic states of the atoms that make up the crystal. Consequently, the coupling between the electron systems of the individual layers of van der Waals (vdW) materials[4] is encoded in their band structure[5,6]. Heterostructures of graphene and hexagonal boron nitride (hBN) are widely used to screen electrons in graphene from the environment, therefore providing high electron mobility[7–9]. Isolation of the two materials is generally assumed over the full energy range, although small changes in the graphene band structure are observed as a function of the stacking angle of graphene and hBN[10–13].

Here we scrutinize the band structure of graphene–hBN heterostructures over a wide energy range to shed light on the interactions in this most widely used vdW system. We apply a series of experimental techniques based on low-energy electron microscopy (LEEM) to assess structural and electronic properties *in situ*, with high lateral resolution[14,15]. First, we study the band structures of graphite and bulk hBN as a reference. Combining LEEM-based angle-resolved photoemission spectroscopy (ARPES) and angle-resolved reflected-electron spectroscopy (ARRES)[16], we deduce information on both the occupied and unoccupied bands over an unprecedented energy range. Then we investigate the band evolution in few-layer hBN and show that our ARRES data match very well with *ab initio* calculations. Finally, we turn to stacks of few-layer graphene on hBN to study their electronic coupling over an energy range of ~25 eV. All samples are produced on a conductive silicon substrate using a polymer-free assembly technique[17] to guarantee clean surfaces and graphene–hBN interfaces (see Methods section).

## Results

**Band structure of the bulk materials.** Figure 1a,b show local measurements on mechanically exfoliated graphite and bulk hBN flakes of ~20 nm thickness, respectively. The occupied bands (negative energies in Fig. 1a,b) are measured with ARPES using a helium ultraviolet light source[18], whereas the unoccupied bands (positive energies in Fig. 1a,b) are studied using ARRES[16]. This

novel technique offers high lateral resolution, allowing us to measure band structures on small, exfoliated flakes. Although the optimal lateral resolution of ARRES is ~10 nm (ref. 16), here we integrate over larger areas to improve the signal-to-noise ratio. ARRES uses the fact that the reflectivity of low-energy electrons depends strongly on both their kinetic energy $E_0$ (refs 14,15,19) and their in-plane momentum $k_{||}$ (refs 20,21) (both of which can be precisely tuned in LEEM). In particular, the electron reflection probability for a specific combination of $E_0$ and $k_{||}$ is high when the material studied has a band gap at that energy (red in Fig. 1a,b). The reflection probability is low, in contrast, if $E_0$ and $k_{||}$ coincide with an unoccupied free-electron-like band (blue in Fig. 1a,b). The special case $k_{||} = 0$ (normal electron incidence) is regularly used in so-called LEEM-IV experiments[22,23]. It is noteworthy that we cannot measure between the Fermi level (maximum for ARPES) and the vacuum energy (minimum for ARRES). The resulting gap, the work function, is incorporated to scale in Fig. 1. In total, the ARPES–ARRES combination gives insight into an exceptionally wide energy range of the band structure. In fact, in Fig. 1a,b, all bands around the Γ-point are probed, except for the lower edge of the conduction band. Moreover, the combined data are well-described by band structure calculations for bulk graphite and hBN (black lines in Fig. 1a,b, respectively).

Interestingly, ARRES does not only reveal band edge positions. Rather, ARRES probes the full transmission states of a material. Those states can be found by calculating the full electron scattering problem starting from a plane wave in the vacuum half-space above the sample and computing its reflection and transmission at the slab of material (see details in the Methods section and refs 24–26). A calculated ARRES spectrum for bulk hBN is plotted in Fig. 1c and shows striking similarity to the measurements in Fig. 1b. Most intuitively, ARRES data can be compared with the density of unoccupied states projected onto the sample plane. To illustrate this, we calculate the projected density of states for bulk hBN from its Kohn–Sham band structure in the local density approximation[24]. Remarkably, this calculation shown in Fig. 1d does not only describe the large band gaps (observed as red areas in Fig. 1b) but also the subtle pockets in the band structure (for example, the gaps marked with arrows in Fig. 1d). The energy resolution of our LEEM microscope is ~150 meV, which determines the minimal features of the band structure that can be probed in our instrument. It is worth noting

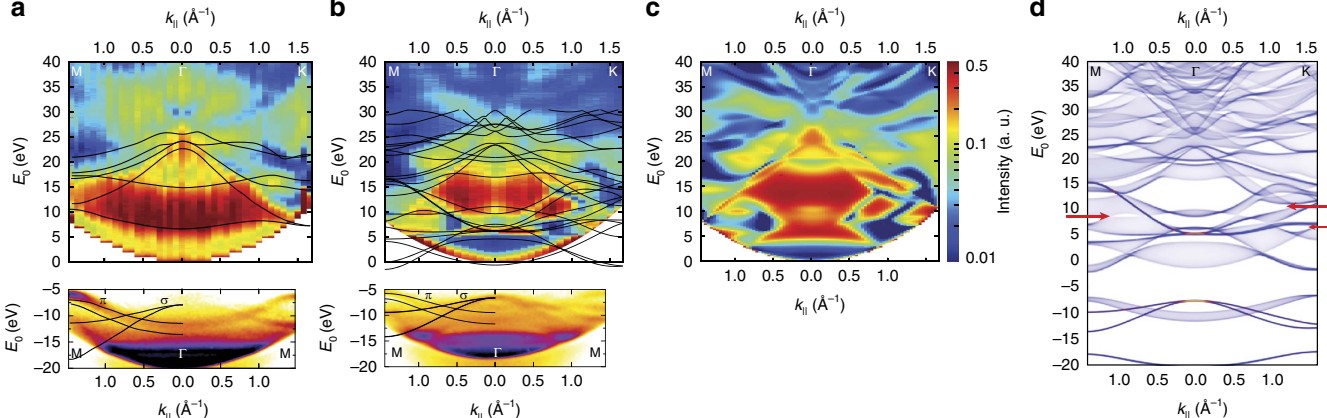

**Figure 1 | Measured band structures of the mother compounds graphite and bulk hBN.** (**a**) Band structure of graphite. The unoccupied bands (positive energies) are measured by ARRES, the occupied bands (negative energies) by ARPES. The full band structure is well described by band structure calculations (black lines, adapted from ref. 27). (**b**) Experimental band structure of bulk hBN measured as in **a**. Black lines are calculated band edges. (**c**) ARRES spectrum calculated with a full-potential linear augmented plane waves method (Methods section and refs 24–26). (**d**) Calculated projected density of states of bulk hBN. The features in the measured unoccupied band structure in **b** can easily be identified with bands and band gaps, as well as fine pockets in the calculations (for example, arrows in **d**).

that the almost perfect match between experimental data in Fig. 1b and theoretical predictions in Fig. 1c,d is achieved by *ab initio* calculations without free parameters. Conversely, ARRES data may serve as a benchmark for more detailed band structure calculations. This is particularly important, as no experimental data have been available in the energy range probed here.

**Band structure of few-layer hBN**. After establishing ARRES as a method to study band properties of graphite and bulk hBN—the mother compounds of vdW heterostructures—let us now discuss these materials in the few-layer limit. It is well known that for few-layer graphene, the continuous conduction band of graphite splits up into quantized transmission resonances. For a stack of $n + 1$ graphene layers, one can find $n$ such transmission resonances, which lead to $n$ characteristic minima in LEEM-IV

curves, thus unambiguously revealing the number of graphene layers[27–29]. These transmission resonances can be viewed in analogy to a tight-binding model where individual transmission resonances correspond to 'atomic' wave functions and are frequently called 'interlayer states'[16,28]. It is noteworthy that in contrast to tight-binding theory, the transmission resonances probed here cannot be assigned to localized states and, in particular, are not spatially localized between adjacent layers. In this tight-binding picture, the energetic splitting of the transmission resonances can be interpreted via a hopping integral that quantifies their interaction. Consequently, the energetic separation of the minima in the IV curves is a direct measure for the hopping integral between the individual 'interlayer states', that is, their coupling. Interestingly, the same arguments apply to few-layer hBN and hence similar discrete transmission resonances are expected[30]. To investigate in detail

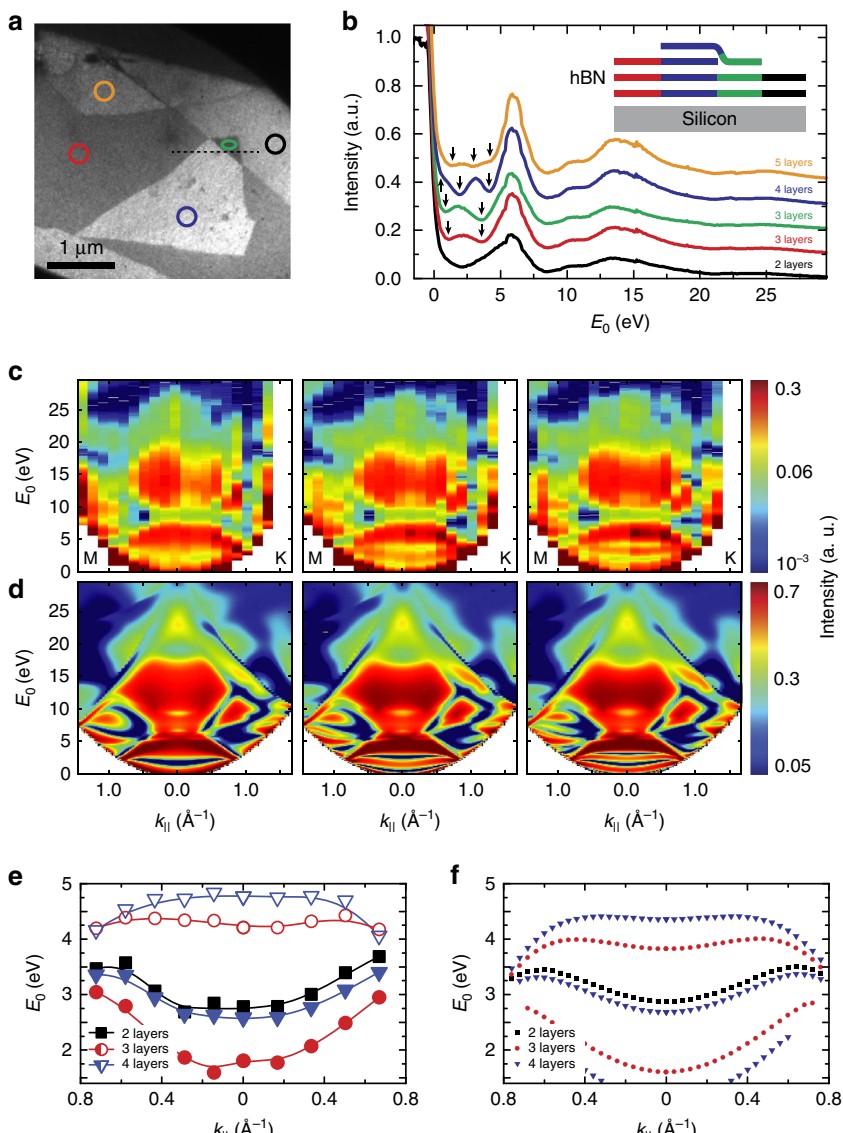

**Figure 2 | Band quantization in few-layer hBN.** (**a**) LEEM micrograph of flakes with different number of hBN layers. $E_0 = 3.7$ eV is used for optimal layer contrast. (**b**) IV curves taken by averaging the intensity of the areas indicated in **a**. Clear minima are observed for $0 < E_0 < 6$ eV. A number of $n$ minima (marked with arrows) corresponds to $n$ quantized interlayer bands and hence, to $n + 1$ hBN layers. The IV curves are vertically shifted by 0.1 a.u. for clarity. The inset shows a sketch of the flake assembly along the line profile in **a**. A monolayer hBN flake lies over a step edge from bilayer to trilayer hBN. (**c**) ARRES measurements performed in the areas of two, three and four hBN layers marked black, red and blue in **a**, respectively. (**d**) ARRES calculations for two, three and four hBN layers. (**e**) Measured dispersion of the quantized interlayer bands extracted from **c**. (**f**) *Ab initio* calculations describe the dispersion well over the full range of measurement.

how the band structure of hBN evolves towards the few-layer layer limit, hBN is mechanically exfoliated and few-layer flakes are selected and deposited onto a silicon substrate. Figure 2a shows a bright-field LEEM image of a few-layer hBN crystal. In particular, we chose an area where a monolayer hBN flake lies over a natural step edge from trilayer to bilayer hBN as sketched in Fig. 2b. This layer assignment can be made when looking at the LEEM-IV curves in Fig. 2b that are taken at the marked areas in Fig. 2a. Indeed, we observe $n$ minima (marked with arrows in Fig. 2b) for $n + 1$ hBN layers, as in few-layer graphene. Clearly, the top monolayer flake adds another quantized transmission state (visible as an additional minimum in the IV curves) to both the three and four hBN layer areas. This indicates that the monolayer flake is electronically coupled to the underlying hBN in the same way these hBN layers are coupled to each other. It is worth noting that the minima expected for $E_0 < 2\,\mathrm{eV}$ are convoluted with the steep mirror-mode transition and are therefore hard to identify or are completely invisible.

Figure 2c shows ARRES measurements performed on bilayer, trilayer and four-layer areas (the size of the used areas is marked by black, red and blue circles in Fig. 2a, respectively). For high energies, these ARRES maps clearly show features identical to bulk hBN (Fig. 1b), confirming that the flakes are few-layer hBN indeed. In addition, we perform *ab initio* ARRES calculations, that is, solving the full electron scattering problem in the vacuum and crystal half spaces (see Methods). The calculations, shown in Fig. 2d, reproduce our experimental data in almost all details for higher energies and for $0\,\mathrm{eV} < E_0 < 5\,\mathrm{eV}$ as well. The latter is the most exciting part, being the energy range of the quantized quasi-two-dimensional bands of bilayer, trilayer and four-layer hBN. Their respective dispersion relations are deduced from both Fig. 2c (experiment) and Fig. 2d (theory), and are plotted in Fig. 2e,f, respectively. The correspondence between theory (Fig. 2d,f) and experiment (Fig. 2c,e) is excellent without free parameters, demonstrating the strength of ARRES in getting quantitative data on unoccupied bands and their evolution towards the few-layer limit.

**Studying the coupling within heterojunctions.** From Fig. 2, we conclude that transmission resonances exist in the same energy range for both few-layer hBN and graphene[16,27,28]. Consequently, one may expect the corresponding electronic states to interact strongly once the two materials are joined together, especially as the stacking distance is comparable to the interlayer spacing in bulk hBN and graphite. Let us suppose this were true. Then, on stacking few-layer graphene on bulk hBN, the graphene interlayer states should mix with the broad hBN bulk band to form one continuous band. In that case, LEEM-IV curves should exhibit a broad minimum without oscillations for $0\,\mathrm{eV} < E_0 < 5\,\mathrm{eV}$. However, this scenario is in stark contrast to the experimental results. In Fig. 3b, we show IV curves for several thicknesses of graphene, stacked onto bulk hBN (see Fig. 3a,c). Remarkably, for low energies the IV curves clearly show the oscillations related to the quantization of the graphite band into $n$ transmission resonances for $n + 1$ layers, just similar to that for few-layer graphene on Si. Hence, the intimate contact to the layered hBN crystal below does not lead to a common continuous band. Moreover, if we zoom out to the full energy scale (0–30 eV), we find that the IV curves continuously evolve from hBN-like to graphite-like as the number of graphene layers $n$ increases. Specifically, all hBN-related features (most prominently the deep minimum around $E_0 \approx 8\,\mathrm{eV}$ in Fig. 3b) vanish for $n = 4$. Remarkably, this layer dependence is well-described by a linear combination of the experimental IV curves of graphite, $I_G(E_0)$, and bulk hBN, $I_{\mathrm{hBN}}(E_0)$ (green and blue curves in the first panel

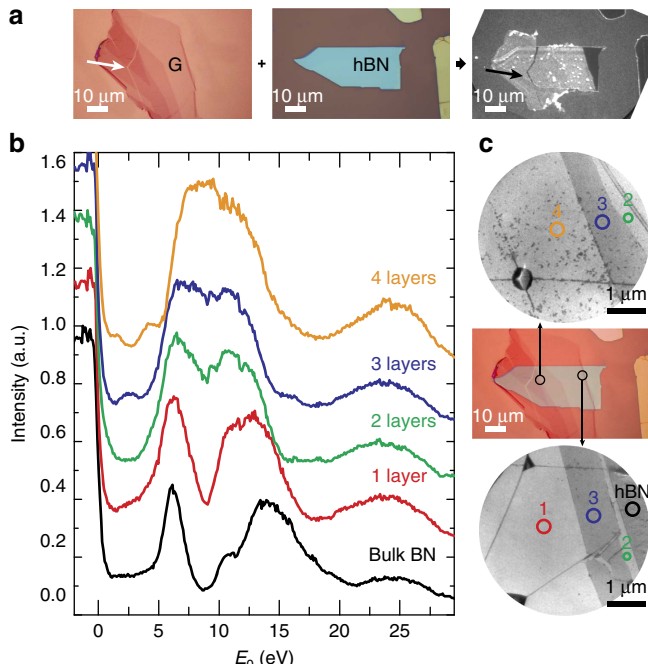

**Figure 3 | Interaction of electronic bands of graphene and hBN.** (**a**) Few-layer graphene is transferred onto a hBN flake (both selected using optical microscopy). A PEEM image (right) of the assembled graphene-hBN stack shows clear contrast between the different materials. (**b**) LEEM-IV curves resolve the thickness-dependent oscillations between $1\,\mathrm{eV} < E_0 < 5\,\mathrm{eV}$ characteristic for few-layer graphene clearly. (**c**) LEEM images (top and bottom) acquired at $E_0 = 4.3\,\mathrm{eV}$ together with an overlay of the two optical images from **a**. Comparison of the three images allows it to identify areas with different numbers of graphene layers. The areas where IV curves are taken are marked. The dark lines in the LEEM images are wrinkles in the graphene sheet.

of Fig. 3d, respectively). In Fig. 4a, we fit a simple Lambert–Beer law (red lines) to the measured IV curves (black circles) for all samples where the total intensity $I_{\mathrm{vdW}}$ reflected from the full stack is given by

$$I_{\mathrm{vdW}}(E_0, n) = I_G(E_0) + I_{\mathrm{hBN}}(E_0) \cdot \sigma \cdot \mathrm{e}^{-n/\delta}. \tag{1}$$

The fit works remarkably well for all $n$ and the full energy range (it is noteworthy that the layer-dependent low-energy minima cannot be reproduced with this approach). We find a characteristic penetration depth of $\delta = 0.9$ graphene layers and a ratio of interaction cross-sections of low-energy electrons with graphene and hBN of $\sigma = 15$. Moreover, this simple fit describes the data at least as well as more sophisticated electron reflection calculations that take band structures and coherent effects into account (light blue lines). The last panel in Fig. 4a shows *ab initio* calculations of IV curves for different stacking geometries of graphene on hBN. The result is independent of the stacking geometry, which suggests that the unoccupied bands are not significantly affected by the details of the Moiré superlattice formed due to the lattice mismatch between graphene and hBN, a subtle effect that can be resolved at low temperatures close to the Fermi level[10–13]. Given the success of this simple Lambert–Beer model together with the existence of the quantized graphene bands on hBN, we can rule out interaction of the interlayer states of few-layer graphene and hBN over an energy range of 30 eV, despite the very similar nature and the intimate contact of the two materials (we note that also at much higher energies, no interaction was observed in near edge X-ray absorption fine

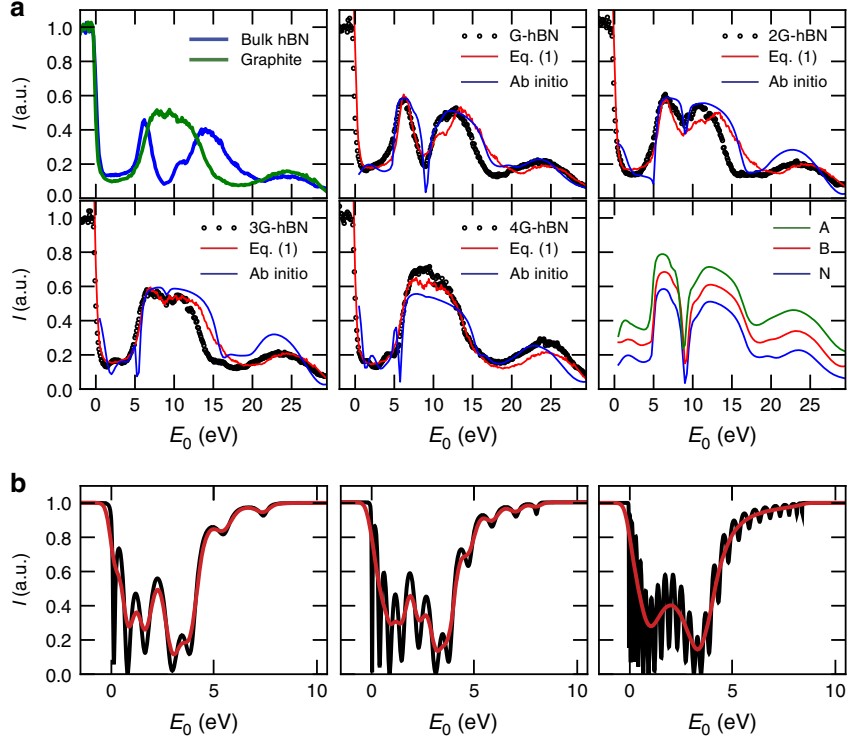

**Figure 4 | Quantitative analysis of the interaction of electronic bands of graphene and hBN.** (**a**) The first panel shows IV curves of bulk graphite and hBN. The next panels compare the data in **b** with two models. Specifically, we follow the hBN-specific minimum at $E_0 \approx 8$ eV that vanishes continuously as the number of graphene layers on top increases. This behaviour is described well by a simple Lambert–Beer law using equation (1) (red) and by *ab initio* calculations (light blue). The last panel shows the *ab initio* result for monolayer graphene on hBN with the carbon atom in the centre of the hBN hexagons (A), above boron atoms (B) and above nitrogen atoms (N). (**b**) Calculated IV curves for three graphene layers on 5, 9 and 22 hBN layers, respectively. Although for five hBN layers a beating pattern obscures the signal, the two minima, which are measured for three graphene layers (*cf.* Fig. 3b) become apparent for a thicker hBN substrate. Black lines are the calculations, red is convoluted with a Gaussian (0.5 eV full width at half maximum) to approximate the experiment.

structure measurements[31]). This is a crucial finding, as the efficient decoupling of the layers is the underlying reason for the high charge-carrier mobility observed in graphene–hBN heterojunctions. It is worth noting that here we chose thin graphene on top of a thick hBN substrate for its relevance as a widely used system for electronic transport studies. The experimental outcome of ARRES measurements in this two-component heterostack is very clear and can be interpreted straightforwardly. In the more complicated case of few-layer graphene on few-layer hBN, for example, a beating pattern between hBN and graphene transmission resonances emerges[32], which bears the risk of obscuring the underlying physics. In these cases, theoretical calculations as shown in Fig. 4b are necessary to understand the IV curves of the combined system. As an example, we show in Fig. 4b how IV curves calculated using a Kronig–Penney model for three layers of graphene on hBN substrates evolve with increasing substrate thickness from 5 to 22 hBN layers. The convergence from a complex beating pattern towards the IV curve of graphene on bulk hBN (as experimentally observed in Fig. 3b) with increasing hBN thickness is apparent in the smoothed calculation (red lines in Fig. 4b). The methodology presented here is thus a versatile tool for studying properties of vdW materials, in particular their interlayer coupling. Combining ARRES with ARPES using a synchrotron source and improving the energy resolution by using an energy-filtered electron source would even allow one to study charge transfer between the layers and determine band offsets in the future.

## Discussion

In this letter, we present a test bed for interactions in vdW materials by measuring quantized transmission resonances in their unoccupied band structure. We show that the conduction bands of few-layer hBN and graphene are quantized in a comparable manner. Despite their similar symmetry and energy range, however, the interlayer states of graphene and hBN do not interact electronically. This is an important insight into the graphene–hBN system, which is the most widely used vdW heterostructure. The methods presented here, specifically ARRES, are directly applicable to more complex vdW materials, for example, stacks including transition-metal dichalcogenides. Consequently, we can study (lack of) electronic overlap between the layers quantitatively. Similar to chemistry, where molecular orbitals are formed from atomic orbitals via their interaction to create compounds with novel properties, this brings us one step closer to a quantitative chemistry of layers. Detailed insight into the interlayer coupling marks an important step towards creating vdW heterostructures with properties not available in conventional materials.

## Methods

**Sample fabrication.** We exfoliate flakes from highly oriented pyrolitic graphite and bulk hBN crystals onto Si/SiO$_2$ substrates. Few-layer flakes are selected via optical microscopy. For the vdW assembly, the top few-layer graphene flake is picked up from the Si/SiO$_2$ substrate with a polydimethylsiloxane stamp coated with an adhesion layer of poly(bisphenol A-carbonate). vdW forces allow us to pick up hBN with this graphene layer. The whole assembly is placed onto a conductive silicon sample via thermal release of the poly(bisphenol A-carbonate) and wet-

chemical cleaning in chloroform and isopropanol, subsequently. Few-layer hBN is transferred onto silicon substrates using the same method. The measurements presented here do not depend on the substrate used, as long as it is conductive enough to prevent charging. The samples are heated in the LEEM to 500 °C, to desorb any residues from the surface.

**Low-energy electron microscopy.** All measurements presented are conducted in the ESCHER setup[33], which has a commercially available, aberration-corrected FE LEEM P90 of Specs GmbH as its centrepiece. Microscopy is performed at $5 \times 10^{-10}$ mbar and 400 °C, to prevent the formation of carbon-based contaminants under the electron beam.

**Theoretical methodology.** All calculations were performed with a full-potential linear augmented plane waves method based on a self-consistent crystal potential obtained within the local density approximation as explained in ref. 24. The *ab initio* reflectivity spectra are obtained with the all-electron Bloch wave-based scattering method described in ref. 25. The modification of this method for stand-alone two-dimensional films of finite thickness was introduced in ref. 26.

**Data availability.** The data that support the findings of this study are available from the corresponding author upon request.

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

## Acknowledgements

We are grateful to Marcel Hesselberth, Daan Boltje and Ruud van Egmond for technical assistance. We thank Charles Kane for fruitful discussions and Kenji Watanabe for supplying the hBN base crystal. This work was supported by the Spanish Ministry of Economy and Competitiveness MINECO (project number FIS2013-48286-C2-1- P) and the Netherlands Organization for Scientific Research (NWO) via an NWO-Groot grant ('ESCHER'), a VIDI grant (680-47-502, S.J.v.d.M.), a VENI grant (680-47-447, J.J.) and by the FOM foundation via the 'Physics in 1D' programme. C.R.D. acknowledges support from NSF grant DMR-1463465.

## Author contributions

J.J. designed the experiment, fabricated the samples and performed experiment and data analysis. A.J.H.v.d.T. developed data acquisition and evaluation software. E.E.K. performed band structure and ARRES calculations. J.B. and C.R.D. assisted with sample fabrication. All authors contributed to data interpretation and writing of the manuscript.

## Additional information

**Competing financial interests:** The authors declare no competing financial interests.

**Publisher's note**: 

