## [Peer Review File · Nature Communications]

Reviewers' comments:

Reviewer #1 (Remarks to the Author):

GENERAL COMMENTS:

The article entitled Quantifying Electronic Band Interactions in Graphene, Hexagonal Boron Nitride and their Heterostructures present a study on the electronic interaction between stacked materials with a focus on graphene and h-BN. This work is an extension of a previous work by the same group entitled Nanoscale measurements of unoccupied band dispersion in few-layer graphene (DOI: 10.1038/ncomms9926). Here, using a combination of ARPES and ARRES techniques developed by the authors, they provide an experimental evidence of the usually assumed non coupling between graphene and h-BN. These interfaces have been investigated over a wide energy range and the peculiarities of the various systems investigated are clearly resolved and systematically supported by ab-initio calculation that fit the experiments. Although it was observed that graphene and h-BN present transmission resonances in the same energy range as graphene, the quantization is preserved and no coupling (continuous band) is evidenced demonstrating experimentally the effective decoupling between the materials. Interestingly, this technique can be adjusted into lab or synchrotron equipment to enable occupied and unoccupied electronic states characterization. Getting full picture of the electronic coupling taking place at the interfaces at this scale would be a major asset for the design and fabrication on nanoelectronic and optoelectronic devices based on vdW heterostructures. This work present interesting features and results and is suited for publication in Nature Communications after a few clarifications.

SPECIFIC COMMENTS:

1. The concept of "chemistry of materials" is unclear, could the authors elaborate or adapt. I have the feeling that the technique helps to monitor and quantify indeed but the chemistry is inherent to the choice of the stacked materials.
2. Did the author carried out ARPES on graphene and h-BN to observe the position of the Fermi level and possible doping or charge transfer?
3. Could the authors specify what is the energy and spatial resolution achieved for these experiments? 10 nm as in reference 8?. Should the energy range for these experiments be limited to the $0 \text{ eV} < E_0 < 5 \text{ eV}$ range, would the resolution be further improved and to which extent?
4. The authors claim that vdW structures could be probed using this technique. It would be interesting to elaborate on the capabilities of the method:
 - a. In line 169: the authors stress that the outcome is so clear only because the experiments are carried out using thin graphene on top of thick h-BN. In that regard to which extent the technique is suited for vdW heterostructures that are intrinsically atomically thin. Moreover, how this technique compares to NEXAFS spectroscopy that can also probe the unoccupied electronic states of the h-BN/graphene heterostructure as published by Sediri et al. Atomically Sharp Interface in an h-BN-epitaxial graphene van der Waals Heterostructure (Scientific Reports | 5:16465 | DOI: 10.1038/srep16465).
 - b. What is the effective depth probed assuming one stack various 2D materials to fabricate a device considering the aforementioned statement.
 - c. Would it be possible to recover the band offsets between at the interface of two materials (or the various interfaces should one stack several materials) and at what resolution?

5. In the methods, could the authors specify on which substrate these experiments can be carried out. Necessarily Si/SiO₂ or the materials can be on metallic substrate or other.

MANUSCRIPT:

The manuscript is well written and the figures are clearly presented. The references should also include NEXAFS based experiments for comparison.

Reviewer #2 (Remarks to the Author):

I think this article presents an interesting, some-what novel experimental technique, and liked the agreement between the theory and the experiment.

One question is what new information has been learnt from this experiment? The first part of the article studies the unoccupied high energy band structure of few layer graphene and hBN, which may as the authors suggest be useful extra information "bench marking" calculations. The second part of the paper is about studying the interlayer interaction in few layer graphene on hBN, and introducing a technique to study the interaction in VdW materials more generally. One problem however is that the energy resolution limited. In particular it is reasonable to expect the strength of the interlayer interaction for graphene on hBN is around 50meV (e.g. the band gaps opened in Ref 2 were ~30meV). However the energy resolution of the current experiment seems to be around 0.5 eV. I admit that previous studies of graphene on hBN have concentrated on the graphene bandstructure at low energies ($|E| < 0.5\text{eV}$), while the current experiment explores different bands (and orbitals) at much higher energies.

A separate question is whether the layer counting for hBN is really "unambiguous" (as it is for few layer graphene, studied in Ref 8 by some of the authors). Perhaps some arrows would help the reader to count the minima Fig 2b. Also, is possible that the top monolayer hBN flake (in the blue and green region of Fig 2a) is misaligned crystallographically from the rest of the stack? Presumably this would affect (suppress) the formation of the "interlayer states".

Response to reviewers' comments:

REVIEWER 1:

We thank Reviewer 1 for his/her positive review.

1. We meant to use ‘chemistry of layers’ as a metaphor for the way interlayer coupling leads to split (‘bonding’ and ‘anti-bonding’) bands. We chose this wording since this effect is very similar to how we understand chemical bond formation, resulting from hybridization of atomic orbitals, within the tight-binding picture. We thank Reviewer 1 for pointing out that our wording was unclear. We therefore rephrased the corresponding paragraphs.
2. We did not perform ARPES on graphene on hBN. It is, however, known from electronic transport measurements performed by some of us [e.g. C.R. Dean *et al.*, Boron nitride substrates for high-quality graphene electronics. *Nature Nanotechnol.* **5**, 722 (2010)] that graphene on hBN is almost charge-neutral. At the moment it is not possible for us to determine the small Fermi level change in ARPES due to the limited energy resolution of our instrument. In the future, if ARRES is combined with ARPES systems with higher energy resolution, e.g. with a synchrotron setup as Reviewer 1 suggests, charge transfer could be studied in detail.
We added a sentence to the outlook referring to that opportunity.
3. The spatial resolution of the ARRES technique is, indeed, ~10nm. Limiting the used energy range would not strongly affect the resolution. In the measurements presented here, we show data averaged over larger areas to optimize the signal-to-noise ratio. In Fig. 2a, for example, the used areas are indicated by circles. We reworded the relevant paragraphs to make that distinction clear.
4. a. We picked the system of thin graphene on bulk hBN for this study because it is the most widely used heterostack for electronic transport measurements. For this system the conclusions are directly clear from the raw ARRES data. The presented technique, however, works as well for more complicated systems provided theoretical support of the sort presented here is given. A combination

of experimental and theoretical analysis as presented here is common practice in other spectroscopy fields e.g. ARPES as well.

We reworded the paragraph to make that more clear. We included a reference to NEXAFS in the revised manuscript.

b. The penetration depth of low-energy electrons in that energy range is $\sim 1\text{nm}$ (the details depend on the exact energy and the materials studied). This is enough for most of the interesting Van der Waals structures.

c. The band offsets between different materials can be probed directly by the presented technique (in the energy range accessible by ARRES) by comparing the shifts of features in the IV-curves of the whole Van der Waals stack with the IV-curves of the individual layers. The energy resolution is $\sim 150\text{meV}$ if the materials studied contain features in the IV-curve that can clearly be identified (as graphene and hBN do). We thank Reviewer 1 for bringing up this interesting future possibility. We included this remark to the outlook section of the manuscript accordingly.

5. These experiments can be performed on any substrate as long as the Van der Waals stack under study is electrically connected to the sample holder to prevent charging. Hence, metallic substrates would be ideal in that respect. We added this statement to the Methods section as suggested.

We thank Reviewer 1 for this remark and have added references accordingly.

REVIEWER 2:

1. We thank Reviewer 2 for his/her valuable comments. As Reviewer 2 points out correctly, we study the band structure far away from the Fermi level. Consequently, we cannot probe the small gap opening that was observed in Ref. 2. For the energy range accessible to ARRES, the interlayer interaction within graphene and hBN respectively is on the order of $\sim 1\text{eV}$ and therefore clearly resolved. A much weaker graphene-hBN interaction could, indeed, remain unnoticed by ARRES. In the future, the energy resolution could be improved to $\sim 10\text{meV}$ by using an energy filtered electron gun. We included this comment into the revised version of the manuscript.

2. The physical mechanism underlying the layer counting scheme – the splitting of a continuous band into individual subbands in the few-layer limit – is identical in hBN and graphene. Consequently, determining the layer number by counting minima is equally robust. For hBN, one additional complication arises, namely that the lowest-energy minima are convoluted with the mirror-mode transition as described in the manuscript. We thank Reviewer 2 for pointing out that the wording was somewhat unclear. We rephrased the paragraph accordingly and added arrows to Fig. 2b as suggested.

3. In Fig. 2a, the crystallographic orientation of the monolayer flake with the underlying layers is not controlled. The formation of the ‘interlayer states’, however, is not affected by this misalignment. This can be seen by the fact that the IV-curves on the 2+1 layer area and the 3 layer area (green and red circles in Fig. 2a, respectively) both show two minima. We added a statement to the manuscript to clarify this fact.

Reviewers' Comments:

Reviewer #1 (Remarks to the Author)

Dear Editor,

The authors have clarified the various points I raised in my first review. The manuscript has been modified accordingly and I therefore have no further request. I would recommend the publication of this work.

Best Regards

Reviewer #3 (Remarks to the Author)

The authors have dealt fairly with all my earlier comments